# Zongertinib, a Novel HER2 Tyrosine Kinase Inhibitor, Maintains an Anticancer Activity for Trastuzumab Deruxtecan-Resistant Cancers Harboring HER2-Overexpression

**DOI:** 10.3390/ijms262110515

**Published:** 2025-10-29

**Authors:** Takashi Kurosaki, Shinichiro Suzuki, Kimio Yonesaka, Yusuke Kawanaka, Toshiyuki Takehara, Takeshi Teramura, Kazuko Sakai, Kazuto Nishio, Hidetoshi Hayashi

**Affiliations:** 1Department of Medical Oncology, Kindai University Faculty of Medicine, Sakai 590-0197, Japan; kurosaki_t@med.kindai.ac.jp (T.K.); s.suzuki-oncologist@med.kindai.ac.jp (S.S.); 1954c0@med.kindai.ac.jp (Y.K.); hidet31@med.kindai.ac.jp (H.H.); 2Division of Cell Biology for Regenerative Medicine, Institute of Advanced Clinical Medicine, Kindai University Faculty of Medicine, Sakai 590-0197, Japan; takehara@med.kindai.ac.jp (T.T.); 077038@med.kindai.ac.jp (T.T.); 3Department of Genome Biology, Kindai University Faculty of Medicine, Sakai 590-0197, Japan; kasakai@med.kindai.ac.jp (K.S.); knishio@med.kindai.ac.jp (K.N.); 4Center for Genomics, Life Science Research Institute, Kindai University, Sakai 590-0197, Japan

**Keywords:** trastuzumab deruxtecan, HER2, tyrosine kinase inhibitor

## Abstract

Trastuzumab deruxtecan (T-DXd), an antibody–drug conjugate comprising trastuzumab linked to a payload DXd, has been extensively used to treat various cancers harboring HER2 overexpression. However, resistance development has been a major challenge to T-DXd treatment. To explore treatment strategies for T-DXd-resistant cancers, we examined two T-DXd-resistant cells named DSR32 and DSR4, which were obtained from HER2 gene-amplified H2170 lung cancer and N87 gastric cancer cells, respectively. The uptake of T-DXd and its transport to lysosomes in DSR32 cells were reduced. Subsequently, *HER2* gene copy number (from 54 to 12) was reduced, which decreased HER2 expression on the cell surface. Thus, T-DXd-resistance might be observed due to the reduced T-DXd uptake caused by decreased HER2 expression in DSR32 cells. In DSR4 cells, no change was observed in HER2 expression and in the uptake and transport of T-DXd. The reduced linker cleavage activity may be associated with T-DXd resistance in DSR4 cells. Meanwhile, both DSR32 and DSR4 cells maintained HER2 activation; thus, zongertinib, a HER2-selective tyrosine kinase inhibitor, blocked the HER2 pathway, induced apoptosis, and inhibited colony formation. Overall, zongertinib can provide therapeutic relief to patients with HER2-overexpressing cancer who have developed resistance to T-DXd.

## 1. Introduction

HER2 is a member of the epidermal growth factor receptor (EGFR) family along with HER1, HER3, and HER4. Dimerization of HER2 with other members of the EGFR family, typically owing to the overexpression of HER2, results in the autophosphorylation or transphosphorylation of tyrosine residues within the cytoplasmic domain of the heterodimer, initiating cell signaling cascades in multiple pathways, such as the mitogen-activated protein kinase and phosphatidylinositol-3-kinase pathways. The activation of these cascades results in cell proliferation, survival, and drug resistance in cancer cells [1,2]. Abnormalities in the *HER2* gene, such as mutations or amplification, act as oncogenic drivers, and a number of HER2-targeted therapies have already been implemented in clinical practice [2,3].

Trastuzumab deruxtecan (T-DXd) is a HER2-targeted antibody–drug conjugate (ADC) composed of an antibody, a cleavable linker, and a topoisomerase I inhibitor payload, DXd [4]. The advantage of this unique structure is widening the therapeutic window of payloads by selectively delivering the ADC to target tumor cells [5]. To date, T-DXd has become the standard therapy for several kinds of malignancies [3,6,7,8,9,10,11]. In HER2-overexpressing breast cancer, T-DXd exhibited an improvement in progression-free survival compared to trastuzumab emtansine (T-DM1), another trastuzumab-based ADC harboring a non-cleavable linker and a maytansine derivative as its payload, in the second-line setting [7,8]. T-DXd also achieved an improvement in survival outcomes of HER2-low breast cancer compared with conventional therapies after progression on one or two prior lines of chemotherapy [9]. Regarding HER2-overexpressing gastric cancer, T-DXd significantly improved response and overall survival in comparison with standard chemotherapies among patients who had progressed during at least two prior therapies, including trastuzumab [10]. In non-small-cell lung cancer, a single-arm, double-cohort phase II trial showed durable anticancer activity of T-DXd in patients with *HER2* mutations who had relapsed following standard therapies, leading to the US Food and Drug Administration approval [11]. The promising results of HER2-overexpressing cohort support further investigation of T-DXd in this setting [12].

However, most advanced-stage malignancies treated with T-DXd remain incurable because of the development of resistance or a lack of initial response. Thus, exploring the mechanisms of resistance to T-DXd and developing strategies to overcome them are urgently needed. Recently, zongertinib, a novel HER2 tyrosine kinase inhibitor (TKI), has been reported to show a promising response in solid tumors harboring *HER2* alterations in a phase Ia study [13]. A key advantage of zongertinib is its ability to circumvent EGFR inhibition due to its high selectivity for HER2, thereby reducing associated toxicities and providing a superior anticancer effect compared with that of conventional HER2-TKIs [14]. However, its efficacy against tumors acquiring resistance to T-DXd remains insufficiently explored.

In the present study, we established T-DXd-resistant cells from two HER2-overexpressing tumor cell lines, H2170 lung cancer and N87 gastric cancer cells. We evaluated the effectiveness of zongertinib against T-DXd-resistant cells in addition to exploring their resistance mechanisms.

## 2. Results

### 2.1. T-DXd-Resistance and Its Relationship to Internalization and Lysosomal Transport

We established T-DXd-resistant cell lines derived from the H2170 lung cancer and N87 gastric cancer cell lines by exposing them to increasing concentrations of T-DXd (1 ng/mL–1 μg/mL) for up to 9 months, as described previously [15,16]. From the pooled T-DXd-resistant cells of H2170 and N87, multiple clones were isolated, and a single clone for each (designated DSR32 and DSR4) was used in subsequent experiments. A growth inhibition assay showed that T-DXd decreased the viability of parent H2170 and N87 cells in a dose-dependent manner, whereas DSR32 and DSR4 cells maintained high viability even at high T-DXd concentrations (Figure 1A,B). The half-maximal inhibitory concentration (IC_50_) of DSR32 cells was higher than that of the parent H2170 cells (>10,000 ng/mL vs. 33.1 ng/mL, respectively; Figure 1A). Similarly, DSR4 cells showed a greater IC_50_ than that of the parental N87 cells (8383.0 vs. 75.7 ng/mL, respectively; Figure 1B). Apoptosis was assessed by detecting cleaved PARP [17] and cleaved caspase-3 [18] using Western blot (Figure 1C,D). Treatment with 1 μg/mL T-DXd induced PARP and caspase-3 cleavage in parent cells; however, these effects were not observed in DSR32 and DSR4 cells.

The anticancer effects of T-DXd depend on its internalization and transport to lysosomes, so we evaluated these processes using pH-sensitive red fluorescent dye–labeled T-DXd. In DSR32 cells, intracellular uptake and delivery of T-DXd to lysosomes were significantly reduced compared with those in the parental H2170 cells (two-way ANOVA, *p* < 0.0001; Figure 1E). In contrast, T-DXd delivery to lysosomes in DSR4 cells was not significantly different from that in the parental N87 cells (*p* = 0.25; Figure 1F). These results suggest that T-DXd resistance in DSR32 cells arises from impaired drug-induced apoptosis, potentially caused by the inhibition of T-DXd internalization and transport to lysosomes (Figure 1G).

### 2.2. HER2 Gene Copy Number Reduction Leads to Impaired T-DXd Internalization in Resistant Cells

Next, we analyzed gene signatures and expression alterations to investigate the causes of T-DXd resistance in DSR32 cells (Figure 2A,B). The significance score for clathrin-mediated endocytosis was not among the top 50, and clathrin gene (*CLTA*, *CLHC1*) expression showed no significant decrease (Figure 2A,B). The expression of caveolin gene (*CAV1*), which is involved in the formation and maintenance of caveolae related to endocytosis, was increased (Figure 2B). No significant changes were observed in the expression levels of the cathepsin B gene (*CTSB*), the cleavage enzyme for the T-DXd linker. Regarding genes associated with the efficacy of DXd, the payload of T-DXd, no variation in the expression level of its target topoisomerase 1 gene (*TOP1*) was observed in resistant DSR32 cells. Furthermore, no significant differences were detected between these cells in the expression levels of ATP-binding cassette transporters (such as *ABCB1* and *ABCG2*), which are potentially involved in the efflux of DXd from cells [19] (Figure 2B, Appendix A).

The signature score for the RAF/MAP kinase cascade was the eighth highest among 1264 signatures (Figure 2A). Next-generation sequencing (NGS)–based gene panel was performed to explore the gene alterations affecting sensitivity to T-DXd involved in the RAF/MAP kinase cascade. No secondary mutations were detected in T-DXd-resistant DSR32 cells. Notably, the *HER2* gene copy number decreased from 53.9 in the parental H2170 cells to 11.5 in DSR32 cells (Figure 2C). When evaluating HER2 protein levels using whole-cell lysates, HER2 protein levels were found to be reduced in DSR32 cells and another resistant clone DSR26 cells compared with those in H2170 cells (Figure 2D). Furthermore, HER2 expression levels on the cell membrane were evaluated using a flow cytometry. The number of HER2 molecules per cell in DSR32 cells was reduced to approximately 25% of that of the parental H2170 cells (Figure 2E). We evaluated whether HER2 protein reduction correlated with T-DXd resistance not only in vitro but also in vivo using a cell line–xenografted mouse model. In tumors of H2170 cells, T-DXd rapidly induced tumor shrinkage, with all six tumors completely disappearing at 3 weeks after exposure (Figure 2F). In contrast, tumors derived from DSR32 cells continued to grow for 3 weeks after T-DXd exposure (Figure 2G).

Collectively, in T-DXd-resistant DSR32 cells, reduced *HER2* gene copy number led to decreased HER2 expression levels on the cell surface. Consequently, the amount of T-DXd binding to HER2 on the cell surface decreased, and the total amount of T-DXd internalized into the cells also decreased (Figure 1G). This may have been the cause of T-DXd resistance.

### 2.3. Resistance to T-DXd May Not Be Solely Due to Impaired Internalization or Transport to Lysosomes

Next, we investigated the resistance mechanism of T-DXd in DSR4 cells. DSR4 cells showed no reduction in *HER2* gene copy number compared with the parental N87 cells (Figure 3A). Furthermore, HER2 expression levels on the cell surface were comparable between DSR4 cells and the parental N87 cells (Figure 3B). The signature score for the RAF/MAP kinase cascade was the ninth highest among the 998 signatures (Figure 3C), but no new genetic mutations were detected in the NGS-based panel. DSR4 cells harbored single nucleotide variants in the *HER2* gene, such as S413L, F425L, or L436V, while the allele frequency of these variants was comparable to that observed in the parent N87 cells (Appendix A). Regarding endocytosis, there were no significant variations in clathrin-mediated endocytosis signature in DSR4 cells (Figure 3C), and clathrin gene (*CLTA*) expression levels also showed no significant changes (Figure 3D, Appendix A). In contrast, the expression levels of the caveolin gene (*CAV1*) and its protein were significantly higher in the DSR4 cells than those in the parental N87 cells (Figure 3D, Appendix A). However, the transport level of T-DXd to lysosomes was greater in the DSR4 cells than that in the parental N87 cells (Figure 1F).

Regarding genes associated with the efficacy of the payload DXd, no variation was observed in the expression levels of its target *TOP1* and its efflux pumps (*ABCB1* and *ABCG2*) between these cells (Figure 3D). Additionally, we performed a growth inhibition assay of DXd to compare its sensitivity between N87 and DSR4 cells. Both cell lines showed decreased viability in a dose-dependent manner with comparable IC_50_ values for the parental N87 and DSR4 cells (105.4 and 45.3 nM, respectively; Figure 3E).

Given that HER2 expression was maintained in T-DXd-resistant DSR4 cells, we hypothesized that another HER2-targeted ADC, T-DM1, may be effective against them. That is because T-DM1 uses the same antibody as T-DXd, trastuzumab; however, it has a different linker and payload system. Growth inhibition assays showed that T-DM1 reduced cell viability in T-DXd-resistant DSR4 cells similarly to parental N87 cells in a dose-dependent manner (IC_50_: 2.7 and 4.1 ng/mL, respectively; Figure 3F). Additionally, a colony-formation assay was performed to evaluate the effects of long-term T-DM1 exposure. Ten days of exposure to either 1 μg/mL T-DM1, almost completely inhibited colony formation in both the parental N87 and resistant DSR4 cells (Figure 3G).

These results suggest that the cause of T-DXd resistance in DSR4 cells may lie not in the impaired internalization or transport to lysosomes, but rather in the linker cleavage process within lysosomes (Figure 1G).

### 2.4. T-DXd-Resistant Cells Demonstrated Susceptibility to Zongertinib

Next, we evaluated the activity of HER2 and its downstream pathways, AKT and ERK, following the acquisition of T-DXd resistance. Both DSR32 and DSR4 cells maintained a phosphorylation of HER2 and its downstream AKT and ERK even after developing resistance (Figure 4A,B). Exposure to zongertinib, an irreversible HER2-selective TKI, resulted in the significant decrease of HER2 and its downstream AKT and ERK phosphorylation in both DSR32 and DSR4 cells as well as in their parent cells (Figure 4A,B). Then, a cell growth inhibition assay was performed to evaluate sensitivity to zongertinib for 3 days. Zongertinib reduced cell viability of parent H2170 and N87 cells in a dose-dependent manner (Figure 4C,D). Furthermore, zongertinib reduced cell viability of both T-DXd-resistant DSR32 and DSR4 cells to the same degree as their parent cells (Figure 4C,D). Their IC50 values are 69.0, 22.2, 3.0, and 6.7 nmol/L for H2170, DSR32, N87, and DSR4 cells, respectively. Furthermore, to assess apoptosis caused by zongertinib, Western blot was performed for evaluating cleaved caspase 3 and cleaved PARP using these cells exposed to 100 nmol/L zongertinib for 24 h. Zongertinib induced apoptosis equivalent to that of the parental cells in resistant DSR32 and DSR4 cells (Figure 4E,F). The effects of extended zongertinib exposure were additionally evaluated using a colony formation assay. In parent H2170 and N87 cells, zongertinib caused colony formation inhibition comparable to that of T-DXd (Figure 4G,H). In DSR32 cells, colony formation was not inhibited by exposure to T-DXd, but zongertinib was able to inhibit it to 30.0% of the control (Figure 4G). In DSR4 cells, approximately 45.1% colony formation was observed even after T-DXd exposure, but zongertinib strongly inhibited this (Figure 4H).

Based on the above, conceivably even T-DXd-resistant cells might depend on HER2 signaling for their survival. Although the degree varied among cells, zongertinib may induce cancer cell death even in T-DXd-resistant cells by inhibiting the HER2/AKT/ERK signaling pathway.

## 3. Discussion

In this study, we established resistant clones from two HER2-overexpressing cancer cell lines. Although exploratory, our analyses suggested that reduced HER2 expression in one clone and impaired linker cleavage in the other may underlie the resistance mechanisms. Even after developing T-DXd resistance, these cells maintained HER2 expression and downstream signaling activity essential for their survival, thereby retaining sensitivity to the irreversible HER2-TKI zongertinib. Given the lack of established treatment options for T-DXd-resistant cancers, the findings of the present study provide valuable insights that may guide future therapeutic strategies for T-DXd-resistant cancers.

The mechanism of action of ADCs is complex and involves antibody binding to the target antigen, internalization, intracellular trafficking to lysosomes, cleavage of ADC, and cytotoxicity mediated by the released payload (Figure 1G) [5,20,21]. This complexity may promote the diversification of resistance mechanisms to ADCs [22]. We investigated which stage of the mechanism of action was impaired in the two established resistant cancer cells. First, we evaluated the internalization of T-DXd into cells and its transport to lysosomes, revealing that this process is impaired in DSR32. While multiple factors can cause ADC internalization defects [22,23], in this case, it is conceivable that the primary cause is reduced HER2 expression on the cell surface, which depends on *HER2* gene copy number. The efficacy of trastuzumab-based ADCs depends on the expression level of HER2 in tumor cells [24,25]. The loss of HER2, particularly the extracellular domain, has been predominantly observed in cancer cells resistant to non-ADC anti-HER2 antibody therapeutics because of reduced dependence on HER2 signaling and weakened immunological effects such as impaired antibody-dependent cellular cytotoxicity [26]. However, in T-DXd-resistant cells, reduced uptake levels of T-DXd may exert a significant impact on resistance [26]. Indeed, numerous cancer cell lines that developed T-DM1 resistance through prolonged drug exposure showed diminished HER2 expression compared with that in their parent cells [27,28,29]. In terms of clinical settings, the DAISY trial, a prospective study evaluating the efficacy of T-DXd across different levels of HER2 expression in metastatic breast cancer, reported that HER2 status evaluated by IHC was reduced at the time of resistance acquisition in 65% of patients [30]. Currently, T-DXd is clinically employed as a therapeutic agent for HER2-overexpressing tumors across multiple cancer types [31,32]. A reduction in T-DXd internalization due to decreased HER2 expression levels may contribute to T-DXd resistance in various cancer types.

Unlike DSR32 cells, DSR4 cells do not exhibit reduced HER2 expression as a mechanism of resistance. However, in clinical practice, resistance to T-DXd has been observed in some patients with preserved HER2 expressions in tumor tissue. This is supported by the aforementioned DAISY trial, in which 35% of cases maintained HER2 expression even after T-DXd resistance [33], and another study reviewed tumor tissues of paired pre- and post-T-DXd treatment showed that HER2 status remained unchanged or was increased in 38% of patients [33]. These clinical findings underscore the significance of the current preclinical study. DSR4 cells had T-DXd internalization levels similar to those of the parental cells and remained highly susceptible to DXd, the payload component of T-DXd, without obvious resistance factors such as increased expression of ABC transporters or loss of TOP1. Although we did not perform direct assays for drug-efflux measurements, this result suggests that the cause of resistance in DSR4 cells lie in the process from lysosomal transport to cleavage of T-DXd. Specifically, T-DXd employs a cleavable tetrapeptide linker, which has been shown to be degraded by certain lysosomal enzymes such as cathepsin B and L [4,34]. In this study, there were no fluctuations in the expression levels of cathepsin B or L in DSR4 cells; however, considering the presence of other cathepsin members and factors affecting cleavage, an evaluation of cleavage efficiency may be necessary. On the other hand, T-DM1 utilizes a non-cleavable linker; thus, its antibody component undergoes proteolytic degradation in the lysosome, generating lysine-conjugated DM1 as the major active metabolite [35]. Therefore, T-DM1 may serve as a useful treatment option for T-DXd-resistant tumors.

This study demonstrated that T-DXd-resistant cells retained sensitivity to the HER2-TKI zongertinib in vitro. Although DSR32 cells exhibited reduced HER2 expression compared with parental cells, they maintained higher *HER2* copy numbers than those in normal human dermal fibroblast cells. Intracellular signaling analysis further confirmed that DSR32 cell survival remained dependent on HER2 signaling. To date, several HER2-directed TKIs have been introduced into clinical practice for patients with HER2-positive breast cancer; however, their efficacy has been modest. The effectiveness of HER2-TKIs is often limited by dose-limiting toxicity resulting from the potential inhibition of other kinases. In particular, diarrhea and skin rash are major concerns, often hindering long-term treatment continuation [36,37]. In the present study, IC50 values for zongertinib were estimated to be below a clinically achievable plasma concentration of zongertinib (Figure 4C,D) [38]. A recent dose-expansion phase Ia study investigated the safety and efficacy of zongertinib in patients with *HER2*-altered solid tumors refractory to standard therapies. The objective response rate was 30.5%, and the most common treatment-related adverse events (any grade/grade ≥3) included diarrhea (50.5/1.0%), rash (16.2/1.9%), anemia (10.5/0%), decreased appetite (9.6/1%), and increased alanine transaminase (9.6/3.8%), indicating that both efficacy and tolerability were superior to those of conventional HER2-TKIs [13]. Our in vitro study suggests the potential of zongertinib as a treatment option for patients with HER2-overexpressing solid tumors who are resistant to T-DXd, warranting further investigation.

There are certain limitations in the present study. First, this study has not fully elucidated the detailed resistance mechanisms in the current resistant clones. In particular, live-cell imaging, endosomal or lysosomal co-localization studies are needed for further clarifying the resistance mechanism. Second, sensitization or synergy of combining HER2-TKI with T-DXd was not evaluated in this study. Further investigation is needed to determine whether HER2-TKI monotherapy can achieve sufficient therapeutic efficacy against T-DXd-resistant tumors, or whether combination therapy with T-DXd yields superior outcomes. Third, our T-DXd-resistant cells were experimentally generated. Only two types of resistant cell lines were used, each representing a different cancer type. Further translational studies are warranted to evaluate HER2 expression and the efficacy of zongertinib using patient-derived cells following the development of T-DXd resistance.

## 4. Materials and Methods

### 4.1. Cells and Reagents

HER2-expressing NCI-H2170 lung cancer cells and HER2-expressing NCI-N87 gastric cancer cells were obtained from the American Type Culture Collection. To generate T-DXd-resistant cells, H2170 and N87 cells were cultured in the presence of T-DXd with gradually increasing concentrations from 1 ng/mL to a maximum of 1 μg/mL for up to 9 months. The cells were maintained in a humidified atmosphere of 5% CO_2_ at 37 °C in RPMI 1640 medium (FUJIFILM Wako Pure Chemical Corporation, Osaka, Japan) supplemented with 10% fetal bovine serum (FBS) and 1% penicillin–streptomycin. Mycoplasma-free conditions were periodically tested using MycoAlert Mycoplasma Detection Kit (Lonza, Basel, Switzerland), and all experiments were performed with mycoplasma-free cells. All parent and T-DXd-resistant cell lines were authenticated using short tandem repeats profiling within the last 3 years. T-DXd, T-DM1, DXd, and zongertinib were purchased from Daiichi Sankyo Co., Ltd. (Tokyo, Japan), Chugai Pharmaceutical Co., Ltd. (Tokyo, Japan), MedChemExpress (Monmouth Junction, NJ, USA), and Selleck Chemicals (Houston, TX, USA), respectively.

### 4.2. In Vitro Growth Inhibition Assay

Cells were seeded in 96-well flat-bottom plates at a density of 1000 cells per well in RPMI 1640 medium supplemented with 10% FBS. After incubation for 24 h, cells were exposed to various concentrations of T-DXd, DXd, T-DM1, or zongertinib. Following further incubation for 6 days (for T-DXd and T-DM1) or 3 days (for DXd and zongertinib), cell viability was assessed using the CellTiter-Glo luminescence assay (Promega K.K., Tokyo, Japan) according to the manufacturer’s instructions. Luminescence was measured using a microplate reader (ARVO MX 1420; PerkinElmer, Shelton, CT, USA). Luminescence values were expressed as a percentage of those observed in untreated cells, and the IC_50_ for each drug was calculated.

### 4.3. ADC Internalization Detection Assay

pH-sensitive red fluorescent dye was conjugated to T-DXd using an antibody internalization detection reagent (Acrobiosystems, Newark, DE, USA) according to the manufacturer’s instructions. Briefly, the detection reagent was mixed with T-DXd and incubated for 10 min at room temperature. For the internalization detection assays, 3 × 10^5^ cells per well were seeded in 6-well plates containing RPMI 1640 medium supplemented with 10% FBS. The following day, cells were exposed to labeled T-DXd and harvested at 2, 4, 8, 16, and 24 h after exposure. Harvested cells were washed with phosphate-buffered saline (PBS), and fluorescence intensity was measured with an LSRFortessa X-20 (BD Biosciences, San Jose, CA, USA). Data were analyzed using FlowJo software version 10.7.2 (FlowJo, Ashland, OR, USA).

### 4.4. Cell Surface HER2 Measuring Assay

Quantification of cell surface HER2 was performed using QIFIKIT (Dako, Tokyo, Japan). Each cell line was treated with a saturating concentration of mouse anti-HER2 antibody (clone 9G6) or mouse IgG2a isotype control, followed by staining with an FITC-conjugated anti-mouse IgG (Dako). Each sample was analyzed by LSRFortessa X-20 (BD Biosciences), and the number of binding sites per cell was calculated according to the manufacturer’s instructions.

### 4.5. Colony-Formation Assay

Cells were seeded in 6-well plates at a density of 3.0  ×  10^4^ cells/well. After 24 h, cells were treated with 0.5, 1 µg/mL T-DXd, 1, 10 µg/mL T-DM1, or 100 nM zongertinib. The medium containing each drug was changed every 3 days. After 10 days of incubation, the plates were gently washed with PBS and fixed with a fixation solution (acetic acid/methanol, 1:7) for 5 min. Colonies were stained with 0.5% crystal violet for 3 h at room temperature. The percentage of colony area was automatically calculated using ImageJ 1.52a (National Institutes of Health, Bethesda, MD, USA).

### 4.6. Western Blotting

Cells were seeded at 2 × 10^6^ cells per plate and allowed to grow overnight in RPMI 1640 medium containing 1% FBS before being harvested. Western blotting was performed as described previously [39]. Proteins were probed using antibodies specific for phospho-HER2 (#2243, 1:1000 dilution; Cell Signaling Technology, Danvers, MA, USA), HER2 (#2165, 1:1000 dilution; Cell Signaling Technology), phospho-AKT (#4060, 1:1000 dilution; Cell Signaling Technology), AKT (#4691, 1:1000 dilution; Cell Signaling Technology), phospho-ERK (#4376, 1:1000 dilution; Cell Signaling Technology), ERK (#9102, 1:1000 dilution; Cell Signaling Technology), cPARP (#5625, 1:1000 dilution; Cell Signaling Technology), cCaspase-3 (#9661, 1:1000 dilution; Cell Signaling Technology), and actin (A2066, 1:100 dilution; Sigma-Aldrich, St. Louis, MO, USA). The actin antibody was used as a loading control.

### 4.7. NGS

We evaluated *HER2* gene mutations and copy number using the Oncomine Tumor Mutation Load Assay (Thermo Fisher Scientific, Waltham, MA, USA), a polymerase chain reaction–based NGS assay.

### 4.8. Microarray

The microarray gene expression analysis was performed using the Affymetrix Clariom™ S Human (Thermo Fisher Scientific) and the Affymetrix Transcriptome Analysis Console software version 4.0.3 (Thermo Fisher Scientific) according to the manufacturer’s instructions. All cells were cultured without T-DXd exposure in the culture medium for one week and were then harvested. Gene expression levels and the significance scores from WikiPathways were calculated using Transcriptome Analysis Console Ver4.0 (Thermo Fisher Scientific).

### 4.9. In Vivo Tumor Growth Inhibition Assay

All animal experiments were performed in accordance with the Recommendations for Handling of Laboratory Animals for Biomedical Research compiled by the Committee on Safety and Ethical Handling Regulations for Laboratory Animal Experiments of Kindai University. The study was also reviewed and approved by the Animal Ethics Committee of Kindai University. H2170 and DSR32 cells (5 × 10^6^ cells/mouse) were subcutaneously injected into the right flanks of female BALB/cAJcl-nu/nu mice (CLEA Japan, Tokyo, Japan). Once tumors reached the target volume (0.2 cm^3^), mice were randomly assigned to treatment and control groups. Mice received a single i.p. injection of PBS (200 μL; control) or T-DXd (10 mg/kg body weight in 200 μL PBS). Tumor volumes and mouse body weights were measured twice per week. Mice were sacrificed if tumors became necrotic or grew to a volume of 2.0 cm^3^. Tumor volume was defined as 1/2 × length × width^2^.

### 4.10. Statistical Analysis

The growth inhibition assay was analyzed using the GraphPad Prism 9.0 software (GraphPad, Boston, MA, USA). The curves were fitted using a nonlinear regression model, and IC_50_ values were calculated accordingly. Statistical analysis was performed using the SPSS version 22.0 (IBM Corp., Armonk, NY, USA). Statistical tests included two-sided, unpaired *t*-tests, one-way ANOVA, and two-way ANOVA, with *p* < 0.05 considered statistically significant.

## Figures and Tables

**Figure 1 ijms-26-10515-f001:**
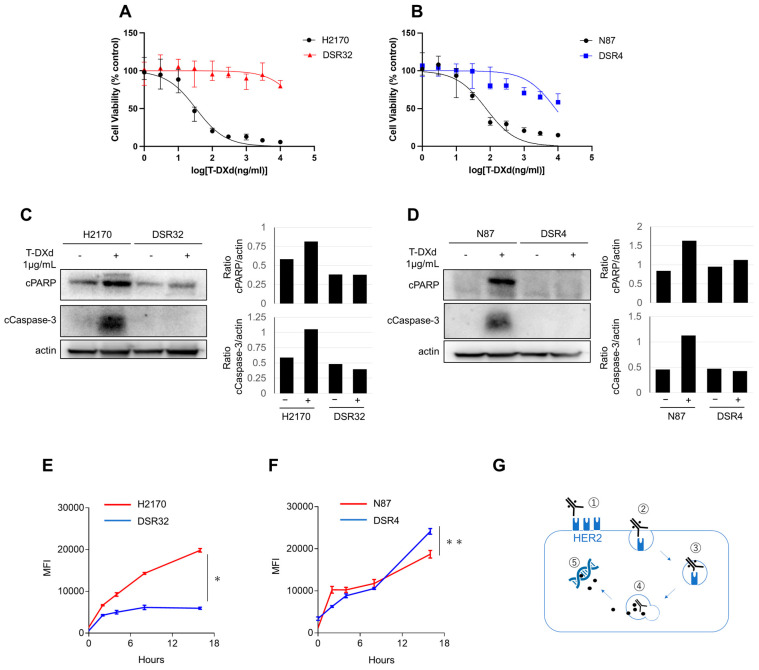
DSR32 and DSR4 cells are resistant to T-DXd. (**A**) HER2-expressing H2170 cells and their clone DSR32 cells were treated with T-DXd at the indicated concentrations. The percentage of viable cells is shown relative to untreated controls (median with a 95% CI of six independent experiments). (**B**) HER2-expressing N87 cells and their clone DSR4 cells were treated with T-DXd at the indicated concentrations. The percentage of viable cells is shown relative to untreated controls (median with a 95% CI of six independent experiments). (**C**,**D**) H2170 and DSR32 cells were treated with T-DXd (1 μg/mL) for 5 days. (**C**) N87 and DSR4 cells were treated with T-DXd (1 μg/mL) for 5 days. (**D**) Proteins were detected by Western blot. The pixel density of the indicated protein was shown as bar graphs relative to actin expression. (**E**,**F**) T-DXd internalization was measured by labeling with pH-sensitive fluorescent dye. H2170 and DSR32 cells (**E**) or N87 and DSR4 cells (**F**) were treated with T-DXd (1 μg/mL) for the indicated times and evaluated at each point. MFI with a 95% CI of three independent experiments is shown. Two-way ANOVA; * *p* < 0.0001, ** *p* = 0.25. (**G**) The schematic diagram of the intracellular signaling pathway associated with T-DXd is shown. 1. Binding to HER2; 2. Internalization; 3. Transport to lysosome; 4. Cleavage of linker in lysosomes; 5. Topoisomerase 1 inhibition.

**Figure 2 ijms-26-10515-f002:**
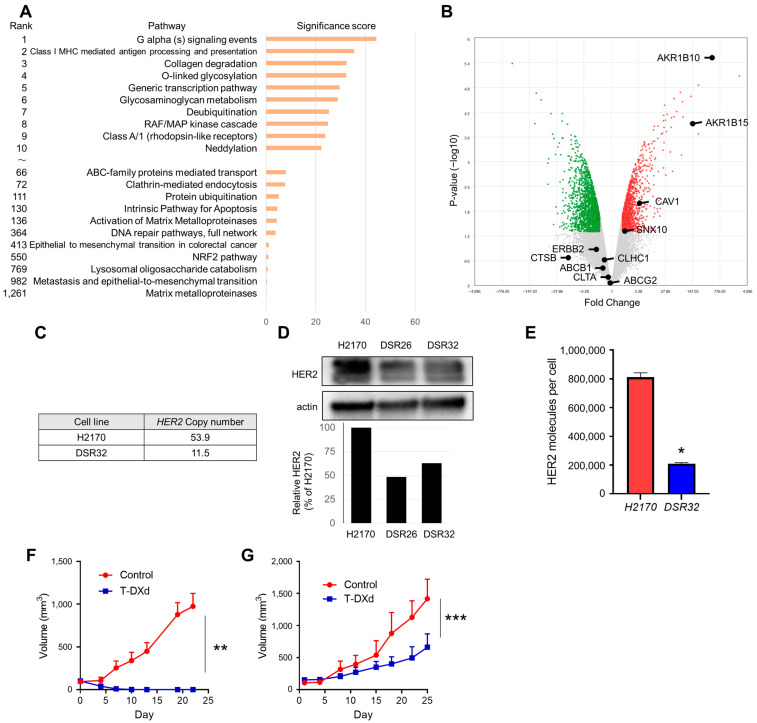
Decreased *HER2* gene copy number in DSR32 cells. (**A**) A ranked pathway with significance score is shown comparing DSR32 vs. H2170 cells. The significance score was calculated by the Affymetrix Transcriptome Analysis Console software version 4.0.3. (**B**) Volcano plot of differential gene expression of DSR32 versus H2170 cells. Each point represents the average value of one transcript in two replicate experiments as revealed by microarray analysis. The X axis represents the linear fold change; Y axis represents −log10 *p*-values from the ANOVA. The expression difference is considered significant for a 2-fold change and a *p*-value of 0.05. Red-colored points indicate significantly enriched genes in DSR32 cells, and green-colored points indicate those in H2170 cells. Gray-colored points indicate no significant difference. (**C**) *HER2* copy numbers of H2170 and DSR32 cells, determined by NGS, are shown. (**D**) HER2 expression of whole-cell lysate of H2170, DSR26, and DSR32 cells was determined by a Western blot assay. The pixel densities of HER2 are shown as bar graphs relative to parental H2170 cells. (**E**) HER2 expression on cell surfaces of H2170 and DSR32 cells, measured by a flow cytometry. Mean HER2 molecule count per cell with the SD of three independent experiments is shown. *t*-test; * *p* < 0.05. (**F**,**G**) In xenograft mouse models of H2170 (**F**) and DSR32 (**G**), tumors were treated with T-DXd (10 mg/kg triweekly) or vehicle. Each treatment group consisted of six mice. Data represent the mean ± SEM. Two-way ANOVA; ** *p* = 0.0002, *** *p* = 0.181.

**Figure 3 ijms-26-10515-f003:**
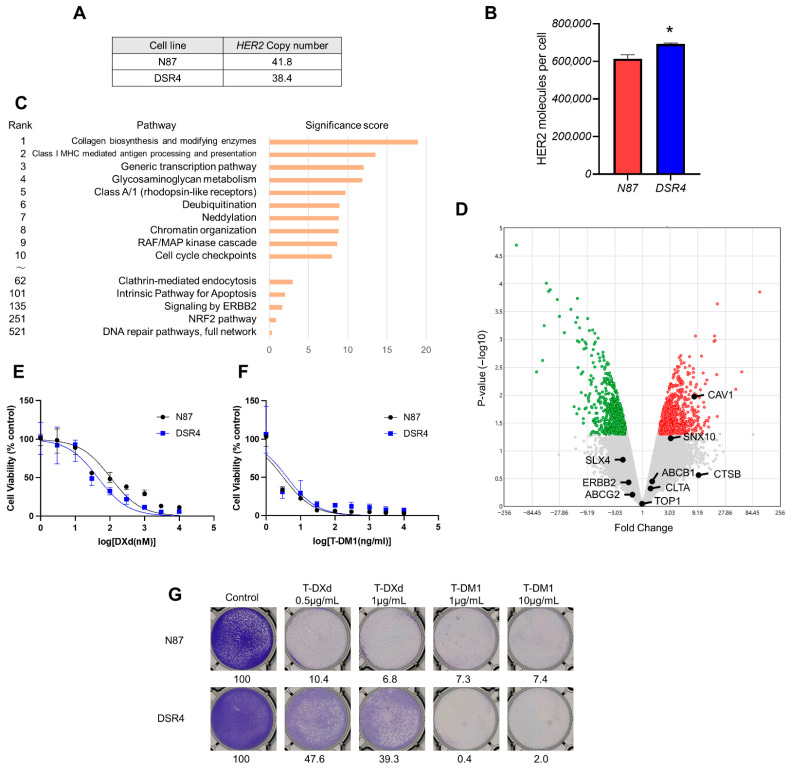
HER2 expression maintained in T-DXd-resistant DSR4 cells. (**A**) *HER2* copy numbers of N87 and DSR4 cells, determined by NGS, are shown. (**B**) HER2 expression on the cell surfaces of N87 and DSR4 cells was measured by flow cytometry. Mean HER2 molecule count per cell with SD of three independent experiments is shown. *t*-test; * *p* = 0.003, (**C**) A ranked pathway with significance scores for DSR4 vs. N87 cells is shown. Significance score was calculated by the Affymetrix Transcriptome Analysis Console software version 4.0.3. (**D**) Volcano plot of differential gene expression of DSR4 versus N87 cells. Each point represents the average value of one transcript in two replicate experiments as revealed by microarray analysis. The X axis represents the linear fold change; Y axis represents −log10 *p*-values from the ANOVA. The expression difference is considered significant for a 2-fold change and a *p*-value of 0.05. Red colored points are significantly enriched genes in DSR4 cells, and green colored points are in N87 cells. Gray-colored points indicate no significant difference. (**E**) N87 and its clone DSR4 cells were treated with various concentrations of DXd. The percentage of viable cells is shown relative to untreated controls (median with 95% CI of six independent experiments). (**F**) N87 and DSR4 cells were treated with varying concentrations of T-DM1. The percentage of viable cells is shown relative to untreated controls (median with 95% CI of six independent experiments). (**G**) Clonogenic assay for N87 and DSR4 cells treated with/without T-DXd, or T-DM1 at indicated concentrations for 10 days The colony-formation area was measured and quantified using ImageJ, and the measurement is shown.

**Figure 4 ijms-26-10515-f004:**
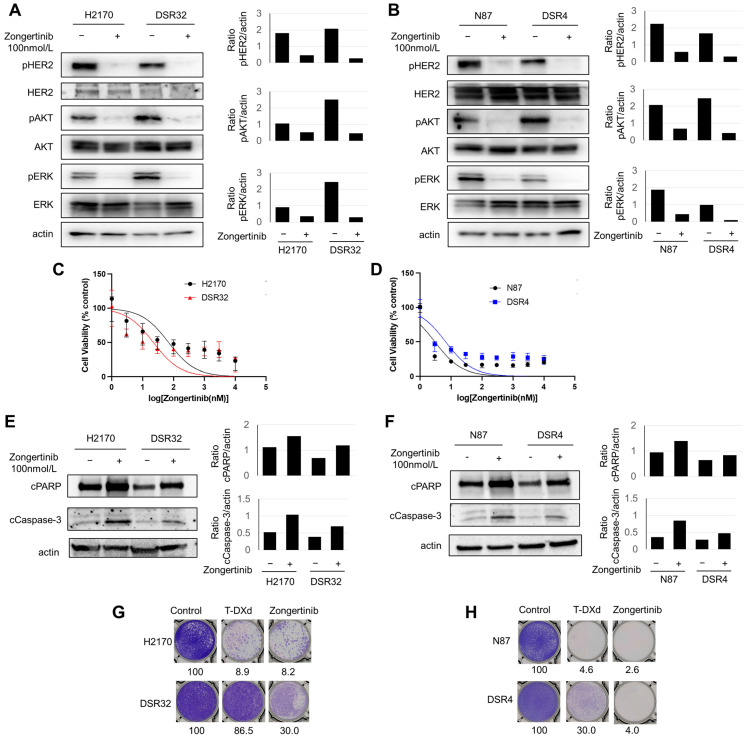
DSR32 and DSR4 cells are susceptible to zongertinib. (**A**,**B**) Either H2170 and DSR32 cells (**A**) or N87 and DSR4 cells (**B**) were treated with zongertinib (100 nmol/L) for 24 h. The indicated proteins were detected by Western blot. The pixel density of the indicated protein was shown as bar graphs relative to actin expression. (**C**,**D**) Either H2170 and DSR32 cells (**C**) or N87 and DSR4 cells (**D**) were treated with zongertinib at indicated concentrations. The percentage of viable cells is shown relative to untreated controls (median with 95% CI of six independent experiments). (**E**,**F**) Either H2170 and DSR32 cells (**E**) or N87 and DSR4 cells (**F**) were treated with zongertinib (100 nmol/L). Indicated proteins were detected by Western blot. The pixel density of the indicated protein was shown as bar graphs relative to actin expression. (**G**,**H**) Clonogenic assay for either H2170 and DSR32 cells (**G**) or N87 and DSR4 cells (**H**) treated with 1 μg/mL T-DXd, 100 nM zongertinib, or vehicle for 10 days. The colony-formation area was measured and quantified using ImageJ, and the measurement is shown.

## Data Availability

The data generated in this study are available upon request from the corresponding author.

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
