# Peer review of "Zongertinib, a Novel HER2 Tyrosine Kinase Inhibitor, Maintains an Anticancer Activity for Trastuzumab Deruxtecan-Resistant Cancers Harboring HER2-Overexpression"

_ijms, 2025, doi:10.3390/ijms262110515_

Round 1
Reviewer 1 Report
Comments and Suggestions for Authors
This study directly identified the resistance mechanism, a typical limitation of targeted therapeutics, and presented an appropriate antibody to overcome this. However, we recommend the following corrections and additions:
Explanation and graph notation for Figure 1E/F
The text does not include an explanation for Panel E in Figure 1. Although an explanation for Panel F exists, it is placed after the complete description of Figure 2, which may cause confusion for readers. The explanations for both panels (E and F) should be presented together in the "T-DXd-resistance and its relationship to internalization and lysosomal transport" section, with a brief recitation of previous results as needed. Also, error bars are not displayed in either graph. Please add them to ensure reliability.
Highlighting and visualizing Western blot/NGS results for resistance mechanism analysis
Western blot and NGS, which aimed to identify the resistance mechanism accurately, are the core of this paper. To enhance reader understanding, we recommend the following: (i) Western blot band images are presented alongside Image J quantification results in bar graphs; (ii) when specific pathways are examined (Fig. 1C and D, Fig. 4A, B, E, and F), a scheme of the intracellular signaling pathway to which the molecule belongs is presented to ultimately determine the functional significance of the observed expression changes; and in NGS fold change graphs (Fig. 2B and 3D), the authors suggest visualizing the relationships between the core molecules highlighted by them in a network format.
Reviewer 2 Report
Comments and Suggestions for Authors
Comments
This work proposes to study T-DXd resistance mechanisms which is an area of great interest in HER2+ and HER2-low cancers, since T-DXd has revolutionized the standard of care for breast cancer and its use has expanded to other solid tumors. This study explores T-DXd resistance and potential alternatives for these cases such as T-DM1.
Major points
- The title states that zongertinib maintains anticancer activity, but throughout the manuscript combination of the TKI and T-DXd was not assayed. No sensitization or synergy with T-DXd is demonstrated.
- The resistance mechanisms described were not mechanistically proven so the conclusions in the different sections should be softened. The data as a whole can suggest hypothesis for each resistance mechanism but as it is presented is more exploratory and descriptive rather than conclusive.
- Statistical details such as p values, CI, R2, n and tests used are inconsistently described in the text and in the figure legends.
- The conclusion of the first section (lines 106-108) should be revised, since the authors show that one resistant model has impaired internalization and the other doesn’t, however state that decreased internalization is not a common mechanism.
- Why was internalization measured after 24h? In HER2 overexpressing cells shorter times are usually assayed.
- There are no statistical comparisons in the graphs presented in the figures, even though the statistical analysis section of Materials and Methods describes tests being used.
- It has been recently reported that is not cathepsin B but cathepsin L the enzyme responsible for linker cleavage of T-DXd (https://doi.org/10.1038/s41467-025-58266-8). Furthermore, a decrease in any CTSB expression could not reflect a reduction in enzyme activity. The same applies to TOP1.
- There is information missing about treatment time and concentration for the microarray experiments.
- Figure 2 and 3 titles should be revised since the information presented is obtained from microarray data and there is no functional assay to establish the resistance mechanism for each model.
- Microarray data or the lists obtained from the analysis are not provided in supplementary materials.
- How were the western blot images quantified? The figure legend states pixel density was used but numbers show relativization.
- DSR32 cells appear to be more sensitive to zongertinib than the parental cell line, how is this explained given that they have less HER2 expression?
- Zongertininb should be tested in vivo alone or in combination with T-DXd to prove that it maintains the anticancer activity of the ADC.
- In DSR32 cells CAV1 expression is increased, however cells show decreased internalization, how is this explained?
- The authors state that DSR32 cells have decreased HER2 expression but have a high copy number compared to normal (lines 335-336), what are normal cells in this case?
Minor points
- English is fluent but there are minor grammar issues that should be corrected for publication.
- Review cited in 3 only addresses breast cancer.
- Line 50: the term HER2-abnormal cancer is not commonly used.
- Line 58: the correct term is non cleavable linker.
- IC50 for DSR32 is not achieved in the graph of Figure 1.
- It should be stated that actin in the western blots was used as a loading control.
- The authors should mention the technique used to study gene signatures and expression.
- Was microarray data validated?
- Graphs should be in a better resolution since some graphs were difficult to see clearly.
- Are volcano plots showing fold change or logFC?
- In the y-axis of volcano plots of Figures 2 and 3 value are misspelled.
- In vivo experiments tested only DSR32 tumors, what about DSR4?
- Some references are redundant.
Round 2
Reviewer 1 Report
Comments and Suggestions for Authors
I appreciate the author's kind response and corrections.